# Nucleic-Acid-Based Molecular Fungal Diagnostics: A Way to a Better Future

**DOI:** 10.3390/diagnostics14050520

**Published:** 2024-02-29

**Authors:** Rajendra Gudisa, Ritika Harchand, Shivaprakash M. Rudramurthy

**Affiliations:** Department of Medical Microbiology, Post Graduate Institute of Medical Education and Research, Chandigarh 160012, India; raaj0047@gmail.com (R.G.); ritz.rajput403@gmail.com (R.H.)

**Keywords:** molecular diagnosis, fungal infection, point of care

## Abstract

The world has seen a tremendous increase in the number of fungal infections during the past two decades. Recently, the World Health Organisation released the pathogen priority list for fungal infections, signifying the importance of these infections in the fields of research and public health. Microbiology laboratories demand an upgrade in the diagnostic system to keep up with the increased burden of these infections. Diagnosis of fungal infections using conventional techniques has always faced limitations in terms of specificity, sensitivity, and turnaround time. Although these methods are the core pillars of the diagnosis, there is an increased need for molecular approaches. Molecular techniques have revolutionised the field of fungal diagnostics. The diverse array of molecular techniques, including techniques like Polymerase Chain Reaction (PCR), have emerged as a cornerstone in fungal diagnostics. Molecular techniques have transformed fungal diagnostics, providing powerful tools for the rapid and accurate identification of pathogens. As these technologies continue to evolve, their integration into routine clinical practice holds the promise of improving patient outcomes through timely and targeted antifungal interventions. This review will cover the molecular approaches involved in fungal diagnostics, moving from the basic techniques to the advanced-level nucleic-acid-based molecular approaches providing a high throughput and decreased turnaround time for the diagnosis of serious fungal infections.

## 1. Introduction

Fungal infections have amplified tremendously over the last two decades, gripping both developing and developed regions. The rise has been attributed to the increased number of immunocompromised patients, immunosuppression, transplants, usage of steroids, and, in recent times, usage of immunomodulatory agents. Besides increasing immunosuppression, fungal infections are also increasingly being reported among patients with severe influenza, COVID-19, admission to ICUs, and patients with solid organ malignancy [1]. Another substantial factor contributing to the increased prevalence is the emergence of antifungal resistance. Approximately 13 million people are infected by fungi annually across the globe, contributing to ~1.5 million deaths [2,3,4]. The most common fungal infections affecting humans include aspergillosis, candidiasis, mucormycosis, cryptococcal meningitis, pneumocystis pneumonia, and neglected tropical infections like sporotrichosis. Of these, mucormycosis is known to cause angioinvasion, infarction, and necrosis, culminating in high mortality. Mortality and morbidity further skyrocket in cases with delay in diagnosis or misdiagnosis, underlining the requisite for timely and precise diagnosis.

Diagnosis of fungal infections is generally established on the combination of clinical, radiological, and laboratory (microbiological/histopathological) findings. Several criteria, like those of the European Organization for Research and Treatment of Cancer and the Mycoses Study Group (EORTC/MSG), Bulpa, Blot, and modified Aspergillosis in ICU (AspICU), have been recommended for timely diagnosis and management of invasive fungal infections in different clinical setups. Conventional modalities have served as the pillars for diagnosis for decades; however, these techniques have inherited bottlenecks of prolonged turnaround time (TAT), the invasive nature of the samples, and lower sensitivity [5]. These include microscopy (KOH/calcofluor-KOH mount), culture on Sabouraud dextrose agar (SDA), and brain heart infusion agar (BHI). The growth on SDA/BHI has to be further identified using lactophenol cotton blue (LPCB) mount or identification by phenotypic techniques or by using matrix assisted laser desorption ionization-time of flight mass spectrometry (MALDI-TOF MS) or API ID cards. Though certain characteristic features have been established for identification using LPCB, this requires precision and experience, especially for rare and novel pathogens. MALDI-TOF MS, with its upgraded database, serves as an effective modality for accurate identification up to the species level via the generation of spectrum profiles. However, these phenotypic methods need the pre-requisite of growth in culture, thereby increasing the TAT. 

Owing to the increasing prevalence of fungal infections associated with high morbidity and mortality, the shift from culture-based to non-culture-based methods has been witnessed. This shift and the use of molecular techniques to delineate the phylogenetic relatedness among fungi have been responsible for taxonomic and nomenclature changes [6]. Non-culture-based diagnostic modalities include nucleic-acid-based molecular techniques, serological, metabolomics, nano-diagnostic approaches, and hybridisation. Serological diagnosis exploiting biomarkers like cryptococcal lateral flow assays, galactomannan (GM), and 1,3, beta-D-glucan (BDG) detection has shown satisfactory results; however, it lacks sensitivity in antifungal therapy for moulds in HIV-negative and non-neutropenic patients [1]. Recently, candidalysin has been identified as a novel biomarker that can help in diagnosing *C. albicans* infections [7,8]. The requisite for the molecular diagnosis of fungi has become the need of the hour, as merely a 12 h delay in the diagnosis of candidemia increases mortality by ~20% [9]. *Aspergillus* infection in immunocompromised patients can be life-threatening, owing to the absence of pathogens in the blood, delaying the diagnosis and culminating in mortality of up to 90% in high-risk patients [10]. The need for molecular modalities is much felt in the absence of or sparse yield of culture or in cases of concurrent infections, wherein it is quite difficult to isolate the fungi in pure form for identification. Molecular diagnosis has the advantage of diagnosing uncommon and unexpected etiological agents, irrespective of the suggestive history and geographical region, as rare pathogens can be missed if a specific diagnostic test is not employed for the diagnosis in the absence of clinical and microbiological suspicion. A similar scenario becomes deadly and life-threatening in outbreaks, as was reported in Joplin, Missouri, wherein thirteen individuals got infected, while five succumbed to illness before the Centre for Disease Control (CDC), Atlanta, could identify *Apophysomyces trapeziformis* as an etiological agent in wound infections, leading to illness [11], with the ability to decipher the lesser-known etiological agents. Besides fungal infections posing a serious threat, antifungal resistance (especially among *Candida* sp.) has convoluted the existing threat to the extent that World Health Organisation (WHO) and CDC have included these under serious threats [12,13].

Molecular diagnostic modalities include techniques for the detection of unique markers in the genetic material (DNA, RNA, or other gene products) of the pathogen. These techniques were initially tedious, labour-intensive, and required much technical expertise; however, recent advancements are focused on overcoming these limitations and streamlining these modalities for utility as a point-of-care test (POCT). An ideal molecular test should have high sensitivity and specificity, with the potential to decipher co-infections and discriminative power to differentiate colonisation from infection in various specimens, viz. blood, body fluids, and tissue, while maintaining cost effectiveness and less technical complexity.

## 2. Sample Collection

Appropriate sample collection is an indispensable component of molecular diagnosis. Adequate sample volume and choosing the right sample for the right infections are significant factors determining the success of molecular modality. The International Fungal PCR Initiative (FPCRI) has evaluated and standardised DNA extraction from varied samples [14]. Owing to challenges faced in diagnosis from blood samples, a minimum of ≥0.5 mL serum/plasma and ≥3 mL whole blood is required for accurate results. Bronchoalveolar lavage fluid (BALF) is the preferred specimen for the detection of respiratory pathogens. However, debrided tissue is the preferred sample in skin and soft tissue infections, invading the mucosal barrier. In cases wherein only formalin-fixed paraffin-embedded (FFPE) tissue samples are available, caution must be taken owing to low DNA yield due to formalin cross-linking and nucleic acid degradation. Moreover, FFPE samples yield low-quality chromatograms in samples with colonisers that interfere with target amplification. FFPE samples are of utmost significance in culture-negative and positive histopathological samples [15,16].

### Challenges Faced in Sample Collection

Owing to a low load of fungi in blood and body fluids, conventional techniques for the detection of fungi do not suffice. Arduous extraction techniques are required for techniques targeting the genetic material and intracellular molecules. Despite advancements in molecular diagnosis, the yield of fungi after extraction methods might be low, partly due to swamping by human nucleic acids. Easily accessible, non-invasive samples have lower yields and usually harbour fungal elements transiently. To overcome these limitations, cell-free DNA is being considered a more suitable target for molecular diagnosis [17,18]. However, mere isolation of the fungus from certain sites like the respiratory tract cannot differentiate colonisers from implicated infectious agents, which might underline the need for isolation of the agent on multiple occasions or determine fungal burden.

## 3. Molecular Diagnosis of Common Fungal Infections

The escalating burden of fungal infections lead to the development of multiple platforms based on varied principles. Broad-range (pan-fungal) or narrow-range (genus or species-specific detection) PCR assays are being used for the direct detection of fungal DNA in clinical samples [19]. Nucleic-acid-based molecular techniques currently available for diagnosing the most common invasive fungal infections, such as pneumocystis pneumonia, invasive aspergillosis, invasive mucormycosis, invasive candidiasis, and invasive diseases due to endemic fungi, are described below.

### 3.1. Pneumocystis Pneumonia (PCP)

Given the non-availability of the culture-based technique, PCR is the key modality of choice for diagnosing PCP, owing to ~95% sensitivity of qPCR [20] compared to microscopic identification. Several studies have been published highlighting the high sensitivity and specificity of PCR compared to microscopy and direct immunofluorescence assays [20,21,22]. A recent meta-analysis has divulged high sensitivity and specificity of ~90% [23], which increased with an increase in fungal burden. Other studies have divulged a positive likelihood ratio of ≥10 and a negative likelihood ratio of ≤0.3 [19]. Owing to high sensitivity and specificity, diagnosis by PCR has been included in consensus guidelines by EORTC/MSG [15]. PCR has been validated and is commercially available for diagnosing PCP from induced sputum, BAL, oropharyngeal wash, and serum/blood, though BALF is the sample of choice with higher sensitivity and specificity. However, the inherent inability of PCR to differentiate colonisation from infection is a matter of concern; hence, the results should be correlated clinically. In order to combat the same limitation, a cut-off value has been established by a few to rule out colonisation [24,25]. Fauchier et al. demonstrated Ct values of 27 and 31 to exclude colonisation, while Ct values of 30 and 25 excluded the infection in (people living with HIV/AIDS) PLHA and HIV-negative patients [26]. However, the specificity of the test decreased from 100% to 80% among HIV-negative patients. Quantitative PCR has also been developed recently to decipher the fungal burden. Sarasombath et al. showed sensitivity, specificity, positive predictive value (PPV), and negative predictive value (NPV) of 100%, 72.9%, 54.9%, and 100% with a Youden index of 0.73, respectively, for a lower cut-off of 1480 copies/μL, in the non-HIV group, PPV, and NPV were noted to be 100%, 91.7%, 97.9% and 100% with a Youden index of 0.92 in the PLHA group, for the upper limit of 9655 copies/μL [27].

Another confounding factor in the evaluation of these tests is the different targets of the test, with different tests targeting mitochondrial small subunit rRNA gene (mtSSU), major surface glycoprotein (MSG), mitochondrial large subunit ribosomal RNA (mtLSU-rRNA) gene, and beta tubulin (BT). Of all these, assays targeting mtSSU were noted to be more sensitive than assays targeting other genes. Furthermore, assays targeting whole nucleic acids (WNAs) were noted to be superior to assays targeting only DNA after intra-laboratory evaluation of five commercials and five in-house assays [28].

The utility of *Pneumocystis* PCR in blood/serum samples has yet to be well established but is of utmost significance, especially in scenarios when invasive respiratory samples are not available. Previous studies have shown comparatively low sensitivity [29], but recent studies have shown promising results when serum PCR is used in conjunction with beta-d-glucan detection. The sensitivity of PCR in serum was 100% in all patients with PCP, while specificity was noted to be 90% in HIV-negative patients (at Ct value of 34) and 100% in blood donors. NPV and PPV were reported as 0.99 and 0.71, respectively. The sensitivity and specificity of positive PCR with BDG of 200 pg/mL were noted to be 92 and 90%, respectively [30].

### 3.2. Invasive Aspergillosis

Since aspergillosis accounts for a substantial burden, diagnostic advances started a long time ago, but the performance of several assays has been variable. Major contributing factors include variable extraction protocols, sample type, sample volume, and different targets. Meta-analysis of PCR for invasive aspergillosis showed an average sensitivity of 80.5%, with a mean specificity of 78.5% for the single result, which rose to 57.9% and 96.2%, respectively, for two different results PPV increased from 36% to 70%, while the NPV was 96% to 94% for one to two tests [31]. Another factor affecting the performance of these assays is the timing of the test, as was revealed by Springer et al., who showed PPV and NPV to vary from 62.5% and 98.3% to 5.4% and 100%, respectively, in patients without and with antifungal *Aspergillus*-active prophylaxis [32]. The target includes multi-copy ribosomal regions with 18S, 28s, and ITS regions.

The European *Aspergillus* PCR Initiative (EAPCRI) Working Group of International Society of Human and Animal Mycology (ISHAM) has validated the methodology for *Aspergillus* PCR in blood, serum, and whole blood [14,33,34,35]. A study by Springer et al. divulged that plasma testing had a sensitivity of 91%, trailed by serum with a sensitivity of 80% and whole blood (55%). In contrast, specificity was highest for whole blood (69%), followed by serum and plasma (69% and 53%, respectively). A recent meta-analysis of *Aspergillus* PCR showed sensitivity and specificity of 79% and 80%, respectively [14], which was comparable to galactomannan detection. Cruciani et al. showed that the sensitivity and specificity of PCR were 79.2% and 79.6% for one positive result, while the same changed to 59.6% and 95.1% for two consecutive results [31].

Besides serum/blood samples, PCR has also been standardised for BAL but has the limitation of an inability to differentiate colonisation from true infection, and PPV was noted to be 72% in the study by Arvanitis and Mylonakis et al. [36]. Meta-analysis on BAL samples revealed sensitivity and specificity to vary from 76.8 to 76.95% and 93.7 to 94.5%, respectively.

Owing to the high specificity of *Aspergillus* PCR in confirming IA, *Aspergillus* PCR is among the EORTC/MSG criteria for defining invasive aspergillosis (IA). In order to be considered as evidence of infection, PCR needs to be positive on two or more consecutive samples, while there needs to be two or more duplicate positive results or one each from BAL and plasma/serum/whole blood [15]. *Aspergillus* PCR from serum has also been included in BII recommendations, while BAL and tissue PCR are part of grade AII recommendations for diagnosis in paediatric patients with cancer or those who have undergone allogenic Hematopoietic stem cell transplantation (HSCT) [36]. Quantitative PCR has also been developed and has been reported to have prognostic value. It has been noted that high levels of DNA (>150 copies/mL) are associated with a higher mortality rate at 90 days. Moreover, patients with negative PCR were noted to have a higher survival rate as compared to patients with positive PCR tests after two weeks of therapy [37].

### 3.3. Mucormycosis

Mucormycosis is considered an emergency life-threatening condition that demands immediate prompt diagnosis to prevent the rapid angioinvasion and necrosis from spreading. Besides facilitating the routine diagnosis, molecular modalities become imperative in concurrent infection with *Aspergillus* and when sparse fragments make the histopathological diagnosis difficult. The lack of serological tests to confirm the diagnosis further underlines the requisite of molecular tests for mucormycosis. The promising targets for *Mucorales* include ITS, rnl, CotH, 18S rDNA, and 28s rDNA. Fresh tissue samples are preferred over FFPE and have been noted to have more sensitivity (86–100% vs. 15–90%). Semi-nested PCR targeting 18S rDNA has shown 81% sensitivity in tissue samples, while PCR-RFLP with ZM1 and ZM3 primers, 18S as targets showed 100% positivity in fresh samples tested against 54% positivity in ITS sequencing. Analytical sensitivity was noted to be 97–100% in fresh tissue samples, while the same was 56–80% in FFPE samples. Modification of PCR with electrospray-ionisation mass spectrometry targeting the 16S–23S rRNA gene has also been employed for diagnosis [38,39,40,41].

For diagnosis of pulmonary mucormycosis, PCR has been standardised in BALf samples as well. Real-time PCR targeting the ITS2 region of rDNA showed 100% sensitivity and 93% specificity [42]. Kasai et al. demonstrated 99% sensitivity and 100% specificity for 28S rDNA from BALf samples [43]. Triple PCR (MucPCR) targeting Acory/Muc1/RMuc showed 100% sensitivity and 97% specificity [44].

Besides tissue and BAL samples, several studies have been conducted to standardise DNA extraction and amplification on non-invasive blood samples. Mucorales DNA in blood has been detected 3–68 days prior to conventional methods, eight days earlier than histology/or culture and three days prior to imaging [45]. PCR targeting 18S rRNA and triple targets (Acory/Muc1/RMuc) have shown variable results. The recent MODIMUCOR study detected *Rhizopus*, *Mucor*, *Rhizomucor*, and *Lichtheimia* in serum samples, with a sensitivity of 85.2% and specificity of 89.8%. The study showed the first PCR result to be positive four days prior to culture collection and one day before imaging. Quantitative PCR was reported to decrease four days after initiation of therapy, with getting negative one week after therapy [46]. Molecular beacon species-specific ITS1/ITS2 region probes for *R. oryzae*, *R. microspores,* and *Mucor* sp. showed positivity in 9/12 samples [47].

Quantitative PCR in serum samples has been noted to be of high prognostic value, and patients with decreasing loads showed better survival rates than patients whose PCR quantitative level remained positive [45]. A commercially available kit, MucorGenius (Pathonostics, Maastricht, The Netherlands), based on RT-PCR, has the potential to detect *Mucorales* in patient sera, with a sensitivity of 75% in 3 h. MycoGenie (Adamtech, Pessac, France) can detect both *Mucorales* and *Aspergillus* with 100% sensitivity, and a study using the kit revealed *Rhizomucor pusillus* as the most common agent [48]. However, more extensive studies are needed to delineate sensitivity and its role in patients with antifungals and rhino-orbito-cerebral mucormycosis. Recent studies using CotH as a target have been conducted on plasma, BAL, and urine, with 100% specificity. Sensitivity was reported to be 90% in urine samples [49]. 

### 3.4. Invasive Candidiasis (IC)

The genus *Candida* comprises a diverse group of species that are commensal inhabitants of the mucous membranes [50]. *Candida* species are usually isolated from non-sterile sites like swabs of gastrointestinal or urogenital sites. However, they do not pose any risk and are of less pathogenic value in asymptomatic individuals with intact immune systems. Invasive candidiasis can have a higher mortality rate in different patient populations, such as patients admitted to the intensive care unit and those on immunosuppressive agents and neutropenic [51]. Conventional culture, such as blood culture, is only positive in less than 60% of patients and has prolonged turnaround time leading to increased mortality rate, as high as 75% [52].

T2Candida is a platform based on magnetic resonance combined with PCR via nanoparticles intermixed with DNA amplicon (ITS2). These magnetic nanoparticle-bound amplicons change the local environment as the amplification proceeds, read as the alteration in the magnetic resonance signal. T2Candida is FDA-approved for use in whole blood, eliminates the need for prior blood culture. It is a very rapid technique which can provide the results within 4–5 h after collection of blood. Analytical sensitivity and specificity have been reported as >91% and >99% [53]. PPV and NPV were noted to be 84% and 99%, respectively, in a setting with a 10% prevalence of candidemia, whereas in low prevalence settings, PPV dropped to 15–31% [53,54]. Meta-analysis delineating its role in diagnosing IC revealed sensitivity and specificity as 91% and 94%, respectively, while sensitivity and specificity rose to 95% and 92% in candidemia [1]. The test has been included in the EORTC/MSG guidelines for diagnosis of IC [15]. However, the testing panel does have limitations in diagnosing all *Candida* species. It can detect *C. albicans*, *C. glabrata*, *C. parapsilosis*, *C. tropicalis*, and *C. krusei*, thereby limiting its role. This limitation has been partly overcome by introducing the T2 *C. auris* panel, which has a limit of detection of <5 CFU/mL for whole blood and skin swabs, with a TAT of 5 h. Furthermore, deploying merely ITS2 as the target compromises specificity; however, owing to the high-performance characteristics and fully automated closed platform, it is of utmost significance in diagnosing IC and candidemia. The commercially available kits and their targets for diagnosis of *Candida*, *Aspergillus,* and *Mucorales* are presented in Table 1.

### 3.5. Endemic Fungi

Endemic mycoses (EM) constitute the diseases caused by varied species of fungi of the genera *Histoplasma*, *Blastomyces*, *Coccidioides*, *Paracoccidioides*, *Talaromyces*, *Sporothrix*, and *Emergomyces* [62]. Diagnosing endemic mycoses has challenged mycologists because conventional techniques like histopathology and culture need expertise and are time-consuming [63]. The serological tests developed for the diagnosis of EM have limitations, which include the dependency of these tests on the host’s immune status and the cross-reactions with other fungi [64].

Specific PCR assays have been developed for diagnosing histoplasmosis using the targets, including ribosomal DNA [65] and ITS 1 and ITS 2 regions [66]. A recent study by Gallo JE et al. has shown the utilisation of predicted protein kinase (PPK) and culture filtrate protein 4 (CFP4) as potential targets for the diagnosis [67].

The molecular methods implemented in diagnosing coccidiomycosis are developed for use in clinical settings and environmental samples [68,69], for certain endemic areas of the United States. In 2018, the US Food and Drug Administration approved using Genestat MDX Coccidioides to diagnose coccidioidomycoses rapidly. A multicentric evaluation has shown a sensitivity of 100% and a specificity range of 94 to 100% [70].

Very few studies have described the use of molecular techniques for the diagnosis of EM, like blastomycosis and talaromycosis, in which the targets used are the BAD 1 [71] gene and mannoprotein 1 (MP1), respectively [72].

## 4. Other Emerging Modalities of Nucleic-Acid-Based Molecular Diagnosis

Isothermal amplification techniques include loop-mediated isothermal amplification (LAMP), nucleic acid sequence-based amplification (NASBA), and rolling circle amplification (RCA). LAMP exploits the thermostable polymerase for displacement of DNA strands and is usually performed at 60–65 °C. NASBA utilises RNA as a template and is performed at 41 °C, while viral polymerase is used in RCA, wherein amplification occurs in a circular form. RCA has the added advantage of target amplification to high copies (10^9^) and is quite tolerant to contamination.

The first report of the diagnosis of *Cryptococcus* infection using NASBA showed that the technique provides good sensitivity and high specificity. LAMP has also been developed for *Aspergillus*, *H. capsulatum*, and *Mucorales*. A recent meta-analysis showed ITS, anzC4, and rRNA as the common targets for LAMP. Specificity was noted to be 87–100% for sputum and 53–100% for BALf and blood samples [73]. However, more detailed studies with large sample sizes are needed. It can also help to monitor the effectiveness of the antifungal treatment [74].

LAMP is sensitive, inexpensive, equipment-free, convenient to use, and requires at least four primers for recognition of six different regions in the target DNA [75]. In the field of mycology, based on the gp43 gene, Endo et al., for the first time, developed a LAMP primer set for *Paracoccidioides brasiliensis* and detected this gene in 22 clinical and seven armadillo-derived isolates [76].

The surge of fungal infections led to the application of sequencing techniques for precise diagnosis. These techniques have the advantage of identifying both known as well as novel species in human infection. The usual target is ribosomes present in multiple copy numbers and loci with multiple conserved ribosomal subunit genes [77]. The preferred targets for sequencing include ITS, small subunit (SSU), large subunit (LSU), Largest subunit of ribosomal polymerase II (RPB1), second largest subunit of ribosomal polymerase II (RPB2) and minichromosome maintenance protein (MCM7). After extensive research, ITS was noted to have a maximum probability of accurate identification, and thereby, in 2012, the International Fungal Barcoding Consortium conferred ITS as a ‘universal fungal barcode’ for fungi [78,79]. When there is no idea of the causative pathogen, PCR amplification of target gene using pan-fungal primers followed by sequencing helps identify the implicated pathogen to the species and subspecies level, both from the clinical specimen or the culture.

In 2014, using NGS, a case was reported for the first time, and since then, more than 300 cases of fungal infections have been elucidated [80]. NGS is of utmost significance in the identification of newer novel agents directly from the sample itself. A study by Irinyl et al. showed the utility of MinION-based NGS for the diagnosis of *Pneumocystis jirovecii* directly from respiratory (sputum and BAL) samples [81]. NGS also helps in differentiating fungal species; for example, *Coccidioides immitis* and *Coccidioides posadasii* can be differentiated, which otherwise are difficult to identify with available conventional techniques [82].

Metagenomic NGS (mNGS) to detect microbial cell-free DNA (mcfDNA) seems to be the light at the end of the tunnel, facilitating the prompt detection of fungal infections, even in the absence of biomarkers. Maximum results of mcfDNA have been generated using Karius^®^ Inc. (Redwood City, CA, USA), which can quantify 1000 pathogens in non-invasive samples like blood as well. A recent study by Hoenigl et al. showed that mcfDNA diagnosed COVID associated pulmonary aspergillosis (CAPA) with a sensitivity of 83% and specificity of 97%. mcfDNA has the potential to identify concurrent infections and unexpected etiological agents in undiagnosed patients, as was noted in a study by Hoenigl et al., wherein reports showed *R. microsporus* in patients suspected to have CAPA [83]. In one of the case reports, authors reached the diagnosis of mediastinal aspergillosis merely within a day of sample collection, which was two weeks before culture and five days before histopathological results [84]. Another study showed *A. lentulus* as the implicated pathogen, which is difficult to differentiate from *A. fumigatus* using conventional modalities but requires different antifungal therapy [85]. A study by Vissichelli et al. divulged the significance of mNGS in altering the antifungal therapy in 12/21 patients tested [86]. Another application is environmental pathogen surveillance, for example, the detection of *C. auris* in wastewater with the use of NGS [87]. Outbreak investigation and clustering of cases can also be investigated to delineate the course of nosocomial transmission [88].

mcfDNA testing has also been exploited in the diagnosis of CNS fungal infections. A prospective multicentric study revealed that mNGS could identify 32/57 infections in comparison to 27/57 by conventional methods [89]. Sometimes, mNGS can also miss the diagnosis of mucormycosis, disseminated aspergillosis, and another angio-invasive disease [89]. Other limitations of mNGS include cost effectiveness for routine laboratories, technical expertise, requisite standardisation, validation, and long turnaround time. Despite all the benefits and future potentials of NGS as described above, implementation into routine diagnostic settings is difficult, especially in resource-limited regions. NGS costs are decreasing but are significantly higher than current diagnostic tools [90]. Moreover, bioinformatics skills are required to process the data for which the right online infrastructure is needed. Presently, the role of NGS is limited to identifying fungal pathogens in patients without a diagnosis, even after performing all the available investigations [1].

## 5. Point-of-Care Molecular Diagnostics

POCT is the need of the hour to attain diagnostic accuracy and to assist stewardship. Though serological POCTs are available, nucleic-acid-based molecular POCTs are still in their infancy. LAMP and Proximity Ligation Assay (PLA)-based assays might serve as the platform for point-of-care diagnosis. PLA provides the combined effect of the sensitivity of real-time PCR and the specificity of the antigen-antibody method of detection. One such example is the development of *Aspergillus* PLA^®^, which showed no cross-reactivity, and positivity was noted in spiked saline and serum samples, as well as BAL samples from three patients. The sensitivity of the assay was noted to be 10–100 times higher than the GM assay [91]. 

Syndromic panels that serve as POCT have been developed for all infectious agents and are a part of the diagnosis of IFDs. The substantial one is BioFire FilmArray Meningitis/Encephalitis panel (bioMérieux, Marcy l’Etoile, France), wherein *Cryptococcus* is included along with other bacterial and viral agents. TAT of the assay is one hour; however, sensitivity has been noted to be low as compared to antigen assays [92]. Fungal pneumonia panels are also available, but none are approved for diagnostic use as of now. A plethora of sepsis panels are available, with the incorporation of *Candida* spp., namely SepsiTest-UMD, SeptiFast RT-PCR, FungiPlex, AusDiagnostics, Sepsis Flow Chip, and T2Candida panel [19]. ePlex systems have combined microfluidics, electrochemical detection, and PCR in an automated platform with a single-use, one-step cartridge-based assay for analysing positive blood cultures and detecting 16 fungal pathogens in addition to bacteria and resistance markers [93]. More extensive studies are needed to determine the role of molecular tests as POCT.

## 6. Role of Molecular Techniques in Determining Antifungal Susceptibility

With the increase in antifungal resistance and increased population of immunocompromised patients, the necessity of antifungal susceptibility testing (AFST) is increasingly used in clinical settings. There are already standardised protocols for the AFST from the Clinical and Laboratory Standards Institute (CLSI) and the European Committee for Antimicrobial Susceptibility Testing (EUCAST) guidelines and some commercially available methods that not only help in the detection of antifungal resistance but also provide guidance towards antifungal therapy [94]. Conventional AFST always had certain limitations, like longer incubation time, suboptimal reproducibility, and non-standardised clinical breakpoints for neglected or overlooked fungal pathogens like dermatophytes. In critically ill patients diagnosed with fungal infections, early detection of antifungal resistance, initiation of appropriate antifungals determines morbidity and mortality [95]. PCR-based commercial kits are available for the identification and detection of the resistant genes. An important breakthrough in the diagnosis is the development of real-time PCR kits, such as AsperGenius^®^, Fungiplex^®^, or MycoGENIE^®^, which can simultaneously diagnose aspergillosis and detect common resistant mutations in *cyp51A* genes such as TR34 and TR36 responsible for triazole resistance. Another important kit that can simultaneously detect Pneumocystis and mutation in the dihydropteroate synthase gene at codons 55 and 57 imparts resistance to a sulfa drug-containing trimethoprim and sulfamethoxazole.

Flow cytometry (FC) and MALDI-TOF MS are promising tools to determine antifungal susceptibility. The results of FC show that this technique is reproducible while testing the AFST of Candida species [96]. MALDI-TOF MS helps in analysing the mass spectra of drug exposed fungal isolates, and the proteome profiles of the control and reference isolates are compared. More refinement of this technique is essential to apply this for routine AFST [97].

## 7. Future Aspects and Challenges

Newer techniques such as clustered regularly interspaced short palindromic repeats (CRISPR)/CRISPR-associated proteins (Cas) system may be a promising future technique for diagnosing fungal infections. Recently, Li, Zhengtu et al. developed a CRISPR-Cas-based diagnostic test for diagnosing pulmonary *Aspergillus fumigatus* infection using the *ITS* region for designing CRISPR Cas RNA (crRNA) and recombinase polymerase amplification (RPA) primer sequence with the T7 promoter [98].

Certain diagnostic methods are raising debate among medical communities in an era where the world is rapidly moving towards molecular diagnostics. Firstly, diagnostic stewardship deals with the appropriate use of these novel techniques, as unnecessary use of these tests can significantly increase the cost of health care. Therefore, these tests should be used with utmost importance, considering various factors, including the immune status of the patient, the type of sample, and the significance of the result in specific population groups [99]. Secondly, from the laboratory perspective, dealing with fungi can have limitations like scarcity of positive controls and contamination [100]. The interpretation of these methods plays an important role as it is very crucial to distinguish between colonisation and infection.

The most significant setback of the current molecular diagnostics of fungal infections is the cost of the tests and affordability itself is major issue in low and middle-income countries. However, we expect that more affordable and reliable molecular tests will be available soon to diagnose fungal infections.

## 8. Conclusions

The current review presents an overview of the molecular methods to diagnose fungal infections. The molecular diagnosis for fungal infections has seen a paradigm shift during the past few years. From molecular markers to Artificial Intelligence, the diagnosis has travelled a long distance, but still, there are miles to cover. With the advent of increasing fungal infections worldwide, the demand for better diagnosis is on the rise, and the scientific world is continuously working towards the same. RT-PCR enables quantification and monitoring of fungal load and has become a crucial tool for assessing disease severity and treatment efficacy. DNA sequencing has played a pivotal role in expanding our understanding of fungal diversity. Sanger sequencing and, more recently, next-generation sequencing (NGS) techniques facilitate the accurate identification of fungal species, even in complex clinical samples. In addition to these single-target approaches, multiplex PCR assays have gained prominence, allowing simultaneous detection of multiple fungal species within a single reaction, accelerating the diagnostic process and reducing costs. From in-house developed PCR to commercial platforms, the technology is advancing. Point-of-care testing incorporating biomarkers and biosensors will provide rapid and easy diagnoses, making bedside diagnosis effortless. The integration of bioinformatics tools is crucial for the interpretation of large-scale sequencing data necessary for accurate species identification, antimicrobial resistance prediction, and epidemiological analyses. Despite the undeniable progress, challenges such as standardisation, relatively high costs, and accessibility remain. Ongoing efforts are needed to overcome these hurdles with emphasis on the need for collaborative research to establish guidelines for implementing molecular techniques in routine diagnostic laboratories.

## Figures and Tables

**Table 1 diagnostics-14-00520-t001:** Some of the commercially available kits and their targets for diagnosing *Candida*, *Aspergillus*, and *Mucorales*.

Name	Manufacturer	Method	Reference
MycoReal *Candida*	Ingenetix, Wien, Austria	Real-time PCR	[55]
SepsiTest-UMD	Molzym Molecular Diagnostics, Bremen, Germany	PCR and Sanger sequencing	[56]
T2Candida	T2 Biosystems, Lexington, MA, USA	Magnetic resonance	[57]
*Aspergillus* spp. ELITe MGB Kit	ELITechGroup S.p.A, Turin, Italy	Quantitative real-time PCR	[58]
MycoReal *Aspergillus*	Ingenetix GmbH, Vienna, Austria	Real-time PCR with melt curve analysis	[59]
*A. fumigatus* Bio-Evolution	Bio-Evolution, Bry-sur-Marne, France	Real-time PCR	[60]
MucorGenius	PathoNostics B.V., Maastricht, The Netherlands	Multiplex real-time PCR	[61]

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
