# Peer review of "Nucleic-Acid-Based Molecular Fungal Diagnostics: A Way to a Better Future"

_diagnostics, 2024, doi:10.3390/diagnostics14050520_

Round 1

Reviewer 1 Report

Comments and Suggestions for Authors

Reviewer’s comment

In the review article titled ‘Molecular Fungal Diagnostics: A way to better future’ the authors Gudisa et al., have described the need for the need of molecular techniques in the diagnosis of fungal infection. Unlike bacteria and viruses, the classification of fungal species is lagging and the advances in molecular techniques help them to reclassify. The authors have covered various aspects of molecular diagnostics in detecting fungal infection and it will be a good collection of information for those interested in fungal diseases. The article is well written. However, the contents can be organized better to have continuity for the readers.  Here are a few suggestions which might increase the value of the manuscript.

1.       To have a better flow, the authors may consider reorganizing the subtitles from line 82 to Line 511. It can be arranged either based on diseases or diagnostic techniques.

2.     The authors may consider including molecular diagnosis for coccidioidomycosis (Ampel et al 2010 PMID: 20948866, McHardy et al 2023 PMID: 36883820)

3.       The authors may consider including Perlin et al 2009 (PMID: 19741524) in the subtitle antifungal resistance. Line 422

4.       Line 12: May consider including ‘rapidity’ as a limitation of traditional diagnostic methods.

Formatting/typo errors

1.       Line 136: ° C

2.       Line 138: 109

3.       Lne, 154, 155 178: Uniformity to be maintained.    Ct vs ct

4.       Line 573:  Uniformity in listing reference    E.g. Reference 16.   Molecular diagnostics in medical mycology Brian L.    Author name to be in the starting of reference.

Comments on the Quality of English Language

The quality of the English language looks fine.  The formatting and typographical errors need to be taken care of.

Author Response

Query: To have a better flow, the authors may consider reorganizing the subtitles from line 82 to Line 511. It can be arranged either based on diseases or diagnostic techniques.

Response :Thank you very much for the suggestions, as suggested we have now completely reorganized the manuscript.

Query: The authors may consider including molecular diagnosis for coccidioidomycosis (Ampel et al 2010 PMID: 20948866, McHardy et al 2023 PMID: 36883820)

Response :The same has been incorporated

Query: The authors may consider including Perlin et al 2009 (PMID: 19741524) in the subtitle antifungal resistance. Line 422

Response :The same has been incorporated

Query: Line 12: May consider including ‘rapidity’ as a limitation of traditional diagnostic methods.

Response :The same has been incorporated

Other Queries

Formatting/typo errors; Line 136: ° C, Line 138: 109, Line, 154, 155 178: Uniformity to be maintained- Ct vs ct, Line 573:  Uniformity in listing reference    E.g. Reference 16 - Molecular diagnostics in medical mycology (Brian L.)    Author name to be in the starting of reference.

Response: We thank the reviewers for the suggestions and all above modifications have been done in the revised manuscript.

Reviewer 2 Report

Comments and Suggestions for Authors

Dear authors,

The present version of the manuscript is a good revision of previous research manuscripts. Some minor errors were identified, which are the following.

Title section: Replace Fungal with fungal.

Lane 136: Replace 650 with 65 C.

Lane 136: Replace 41 0C with 41 C.

Lane 137: The RCA acronym is not described.

Lane 209: The acronyms for EORTC/MSG are not described.

Lane 303: C. albicans must be written in italics.

Lane 329-334: genes' names should be written in italics.

Lane 348: Replace doesn’t with does not.

Table 1. S. no means?

Table 1 in the reference is better described than in the present manuscript. Please update the information with actual references.

Please ensure that all scientific names are written in italics and that all acronyms need to be described.

Comments on the Quality of English Language

Please check all text because some paragraphs include abbreviations (for example, doesn't); the manuscript should be written with scientific language.

Author Response

Queries:

Title section: Replace Fungal with fungal.

Lane 136: Replace 650 with 65 C.

Lane 136: Replace 41 0C with 41 C.

Lane 137: The RCA acronym is not described.

Lane 209: The acronyms for EORTC/MSG are not described.

Lane 303: C. albicans must be written in italics.

Lane 329-334: genes' names should be written in italics.

Lane 348: Replace doesn’t with does not.

Table 1. S. no means?

Table 1 in the reference is better described than in the present manuscript. Please update the information with actual references.

Response: We thank the reviewers for the suggestions and all above modifications have been done in the revised manuscript.

Reviewer 3 Report

Comments and Suggestions for Authors

This manuscript aims to review on the molecular diagnosis of fungi. This is a pretty interesting topic. However, a few important concerns need to be addressed before further consideration. Thus, my suggestion is substantial revision.

Firstly, it’s pretty vague what the authors’ point and logistic is. I understand the rapid diagnosis of pathogenic fungi is important. However, a few essential questions are raised in regards here. Do the authors only try to review species identification? Aside from species identification, there’re still some aspects that the authors should take a look at. For example, the diversity of pathogenic fungi. In C. albicans, the diversity of ECE1 alleles had been reported to contribute to the pathogenicity of C. albicans. Another example, do the authors try to touch upon the fungal detection in different scenario (like from a microbiota) or different phases/cases of infection? This will raise more specific concerns.

Secondly, it’s also vague as to what molecular diagnosis means? Do the authors mean methodologies, such as PCR, Q-PCR, isothermal amplification? There are a number of different such methodologies other than those introduced here. Also, what is the role and aim of sequencing for? It’s not that often (to me) sequencing used to identify fungi in infection cases, but it’s common to study the microbiota. So, what the authors are aiming at? If the authors aim at relatively acute infection, then probably rapid methodologies like PCR or isothermal amplification should be preferrable. If the authors more aim at microbiota (long-term), then sequencing based methods makes more sense.

Comments on the Quality of English Language

The manuscript is well written.

Author Response

Firstly, it’s pretty vague what the authors’ point and logistic is. I understand the rapid diagnosis of pathogenic fungi is important. However, a few essential questions are raised in regards here. Do the authors only try to review species identification? Aside from species identification, there’re still some aspects that the authors should take a look at. For example, the diversity of pathogenic fungi. In C. albicans, the diversity of ECE1 alleles had been reported to contribute to the pathogenicity of C. albicans. Another example, do the authors try to touch upon the fungal detection in different scenario (like from a microbiota) or different phases/cases of infection? This will raise more specific concerns.

Secondly, it’s also vague as to what molecular diagnosis means? Do the authors mean methodologies, such as PCR, Q-PCR, isothermal amplification? There are a number of different such methodologies other than those introduced here. Also, what is the role and aim of sequencing for? It’s not that often (to me) sequencing used to identify fungi in infection cases, but it’s common to study the microbiota. So, what the authors are aiming at? If the authors aim at relatively acute infection, then probably rapid methodologies like PCR or isothermal amplification should be preferrable. If the authors more aim at microbiota (long-term), then sequencing based methods makes more sense.

Response: In the present manuscript, we tried to elaborate on various molecular methods available for diagnosing fungal infections and identifying fungi. Describing the diversity of pathogenic fungi and the methods used to study them will be out of the scope of this review. We have included the different techniques that have been applied to diagnose fungal infections, including their performance. Sequencing from clinical samples has helped in the diagnosis of diseases at an early stage. It can also be used in infections with high clinical suspicion that are microscopy and culture-negative. We agree that Metagenomics NGS is used to study microbiota, and metagenomics to detect cell-free DNA is the latest technique to diagnose fungal infections from clinical specimens.

Round 2

Reviewer 3 Report

Comments and Suggestions for Authors

As responded by the authors, if the authors aimed to review the species identification of fungi, then this manuscript contains pretty limited importance. In terms of fungi diagnosis, novel technologies (NGS, etc.) on species identification may be less important and popular themselves. However, other aspects and concerns in the fungi diagnosis has drawn more attention. For example, the diversity in the alleles of ECE1 of the leading human pathogenic fungi, C. albicans, indicates the pathogenicity. Thus, the product, candidalysin has been studied to be a new target to indicate the infection by this fungi, as the transition from commensal to pathogen. This does not require NGS but only Sanger sequencing.

Comments on the Quality of English Language

The English writing is good.

Author Response

I agree with your views and want to clarify that we only discuss the nucleic acid-based fungal diagnostic, not the biomarker or serological tests. In the revised manuscript, we have explained the use of sequencing-based diagnostics in diagnosing fungal infections directly from the clinical samples. We also have included the detection of candida lysin as a prospect in diagnosing Candida albicans infection.